# Highly indistinguishable and strongly entangled photons from symmetric GaAs quantum dots

Daniel Huber[1], Marcus Reindl[1], Yongheng Huo[1,2,3,4], Huiying Huang[1], Johannes S. Wildmann[1], Oliver G. Schmidt[2], Armando Rastelli[1] & Rinaldo Trotta[1]

The development of scalable sources of non-classical light is fundamental to unlocking the technological potential of quantum photonics. Semiconductor quantum dots are emerging as near-optimal sources of indistinguishable single photons. However, their performance as sources of entangled-photon pairs are still modest compared to parametric down converters. Photons emitted from conventional Stranski–Krastanov InGaAs quantum dots have shown non-optimal levels of entanglement and indistinguishability. For quantum networks, both criteria must be met simultaneously. Here, we show that this is possible with a system that has received limited attention so far: GaAs quantum dots. They can emit triggered polarization-entangled photons with high purity ($g^{(2)}(0) = 0.002 \pm 0.002$), high indistinguishability ($0.93 \pm 0.07$ for 2 ns pulse separation) and high entanglement fidelity ($0.94 \pm 0.01$). Our results show that GaAs might be the material of choice for quantum-dot entanglement sources in future quantum technologies.

[1] Institute of Semiconductor and Solid State Physics, Johannes Kepler University, Altenbergerstraße 69, Linz 4040, Austria. [2] Institute for Integrative Nanosciences, IFW Dresden, Helmholtzstraße 20, Dresden 01069 Germany. [3] Hefei National Laboratory for Physical Sciences at Microscale, University of Science and Technology of China, Hefei, Anhui 230026, China. [4] CAS-Alibaba Quantum Computing Laboratory, USTC Shanghai, Shanghai 201315, China. Correspondence and requests for materials should be addressed to D.H. (email: daniel.huber@jku.at) or to A.R. (email: armando.rastelli@jku.at) or to R.T. (email: rinaldo.trotta@jku.at).

Any prototype of quantum devices that will be used in emerging quantum technologies[1] has to fulfill a long list of requirements. The ideal source of quantum light, for example, should deliver single and entangled photons deterministically, with high purity, high efficiency, high indistinguishability and high degree of entanglement, and it should be compatible with current photonic integration technologies. Implementing such a source, however, is far from an easy task[2,3].

If we restrict the discussion to single photon sources[4], that is if we disregard entanglement, semiconductor quantum dots (QDs) have recently demonstrated their capability to fulfill all the requirements on the wish list, thus carrying great promise for the implementation of photonic quantum networks[1]. This is eventually testified by the increasing interest of the community working with parametric down converters[5–7], the workhorse sources of non-classical light that have been used to achieve a wealth of breakthroughs in quantum optics.

Besides single photons, QDs can also generate triggered polarization-entangled photons during the biexciton–exciton (XX–X) radiative cascade[8]. Having in mind their performances as single photon sources, it would be natural to regard QDs as ideal entanglement sources. The reality, unfortunately, is quite distant from this idea. The main reason is that the development of scalable sources of entangled photons demands approaches that are much more advanced than those employed for single photons. First, the very possibility of using a QD to generate entangled photons poses severe constraints on its structural symmetry[9]. Second, the efficient extraction of both X and XX photons requires sophisticated photonic micro-cavities or broad-band nanowires[10,11]. Third, two-photon resonant excitation schemes are needed to achieve truly on-demand generation of photon pairs and to avoid some of the decoherence processes limiting the indistinguishability of the emitted photons[12,13]. Although the different points have been recently addressed (although never simultaneously on the very same device), non-ideal levels of indistinguishability and degree of entanglement were reported so far[11,12,14,15]. In particular for the latter parameter, all the experiments have shown fidelities to the expected Bell state that rarely exceed 80% (without inefficient postselection techniques)[10,11,14–16], even under resonant pumping[12]. Assuming that the X states involved in the cascade are degenerate[15] and ultra-clean QD samples with reduced charge noise[17] are studied, the degree of entanglement is mainly limited by two effects: recapture processes[11,16] and random magnetic fields produced by the QD nuclei[18,19]. The first one arises when the intermediate X levels are re-excited to the XX level before decaying to the ground state. The second effect is instead related to the hyperfine interaction[20], which couples the spin of the electron and hole forming the exciton with the spins of the QD nuclei, effectively giving rise to a fluctuating magnetic field (Overhauser field[21]) that depolarizes randomly the X states. It is well known that these sources of entanglement degradation lower the maximum entanglement fidelity, and there are strategies to compensate for their deleterious effect, at least in principle.

Recapture processes can be suppressed by shortening the X decay time via the Purcell effect[11] or/and using resonant excitation[12,13]. Nuclear spin-polarization techniques[22] can be used to polarize the nuclear ensemble with the drawback of inducing a finite effective magnetic field, which splits the X states[23]. This splitting can be in turn compensated via additional external perturbations, such as strain, magnetic or electric fields[9,24]. Even if theoretically feasible, this approach is not straightforward to implement experimentally, as the effect of a magnetic field on the X states depends strongly on its direction and can give rise to off-diagonal terms in the X Hamiltonian that cannot be easily compensated with external fixes.

In this work, we look at this problem from a different perspective. Instead of using Stranski–Krastanov InGaAs QDs—the standard choice in QD photonics—we focus on highly symmetric GaAs QDs grown via the droplet etching method[25–28]. In this type of artificial atoms, the effect of the Overhauser field is much weaker than in InGaAs QDs[18]. This is not only due to the absence of indium in the QD nuclei, which has a large nuclear spin of 9/2 compared to gallium with 3/2. GaAs/AlGaAs QDs are indeed free of intrinsic strain fields and composition gradients that have a relevant role on the hyperfine interactions[20]. Moreover, the light- and heavy-hole mixing, an effect which also depolarizes X states[29,30], is minimized to values <5% due to the highly symmetric shape of the investigated QDs[26]. The use of these novel QDs is not the only ingredient to push the degree of entanglement to its limits. Differently from previous attempts on droplet epitaxy GaAs QDs[16], we take advantage of two-photon resonant excitation schemes to coherently populate the XX state[12]. On the one hand, this technique limits the effect of re-excitation. On the other hand, it allows us to demonstrate high visibilities of two-photon interference for consecutive photons emitted by a GaAs QD.

## Results

**Two-photon resonant excitation.** We begin with the characterization of our (GaAs)/AlGaAs QDs, which were obtained by infilling nanoholes created by *in situ* droplet etching of AlGaAs layers (see Fig. 1a–1c and methods)[25,26]. Atomic force microscopy of a representative nanohole (see Fig. 1c) reveals a highly symmetric shape. This shape, which has a strong impact on the QD structural symmetry, is of fundamental importance for the generation of entangled photons. Figure 1d shows a typical microphotoluminescence spectrum of a single GaAs QD under non-resonant excitation. It consists of an isolated X line at high energy and a bunch of neutral and charged multiexcitonic transitions at lower energy (roughly 4 meV below the X). While the detailed discussion of the origin of these states is beyond the scope of this work, we would like to point out that the XX transition is almost always overwhelmed by them and cannot be singled out easily under non-resonant pumping, even at the highest excitation power. This observation is qualitatively ascribed to the large number of confined states in the employed QDs, which lead to many possible charge configurations competing with the XX at high excitation power. In order to make the XX line visible and to gain coherent control over this state, we use a two-photon resonant excitation scheme[13,31,32]. More specifically, we shape the excitation laser into 9 ps pulses (see Methods section) and we tune its energy to half of the energy between the XX and ground state (0) (see inset of Fig. 1e).

The effect of the two-photon resonant excitation in the emission spectrum is shown in Fig. 1e. The multiexcitonic states disappear and the XX line becomes visible with an intensity equal to that of the X. Two additional transitions appear ($C_1$ and $C_2$), which are most probably related to a charged exciton and biexciton. These states, whose detailed investigation is left to future studies, are present in all the QDs we measured. To verify whether the QD is driven resonantly, we measured the power dependence of the intensity of X and XX (see Fig. 2a). Both transitions show well-defined Rabi oscillations[31], thus readily confirming that our excitation scheme allows for coherent control of the XX state. We note that, while the XX state and the crystal ground state are coherently coupled by the laser field, X emission stems from spontaneous decay of the XX state. Therefore the X population reflects that of the XX state and shows Rabi oscillations with the same periodicity. The damping of the Rabi oscillations with increasing laser pulse area most probably arises

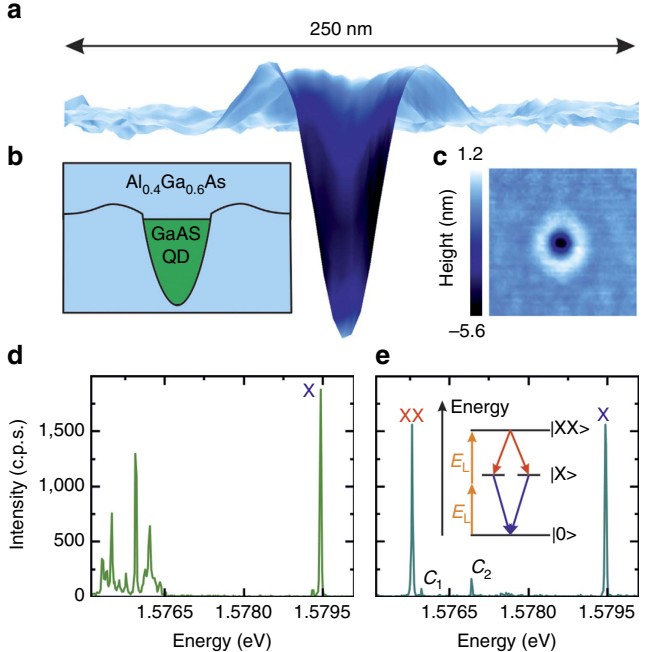

**Figure 1 | Sample structure and spectral properties of the used quantum dots.** (**a**) Cross-sectional 3D view of an atomic force microscopy (AFM) image of a nanohole in an AlGaAs layer. The colour scale reflects the local surface inclination (white for flat areas and black for inclinations >16°). The height-to-width ratio is amplified 17 times to highlight the nanohole shape. A GaAs quantum dot (QD) with a height of about 7 nm is obtained after its filling with GaAs and overgrowth with AlGaAs. (**b**) Sketch of the sample structure. (**c**) Top view of the AFM measurement with colour scale reflecting the local height (0 corresponds to the average height of the flat areas surrounding the nanohole). The lateral scale of the AFM is as in **a** − $250 \times 250\,\mathrm{nm}^2$. The highly symmetric shape is crucial to obtain the high degree of entanglement observed here. (**d**) Microphotoluminescence spectrum of a representative QD under non-resonant excitation (laser photon energy $E_L = 2.54\,\mathrm{eV}$). The neutral exciton emission line is labelled as X. The biexciton line (XX) is not visible in these excitation conditions. (**e**) Spectrum of the same QD as in **d** under resonant two photon excitation ($E_L = 1.5775\,\mathrm{eV}$). The X and XX are visible and have very similar intensity. Two weaker charged states appear ($C_1$ and $C_2$). Inset: For resonant two-photon excitation, the 9 ps laser pulses are tuned to half of the energy difference between |XX> and ground state |0>.

from phonon-induced dephasing[12]. For all subsequent measurements, the power has been set to the π-pulse of the XX.

**Highly indistinguishable photons**. We now characterize the properties of the photons emitted by our QDs in terms of their purity and indistinguishability. In Fig. 2b,c, we report auto-correlation measurements performed on the X and XX lines, which allow us to estimate the value of the second-order correlation function at zero time delay, being $g^{(2)}(0) = 0.007 \pm 0.004$ and $g^{(2)}(0) = 0.002 \pm 0.002$ for X and XX, respectively. This result clearly indicates an extremely high purity of our source.

Next, we study the photon indistinguishability of consecutive photons emitted by our QDs by performing a Hong–Ou–Mandel-type two-photon interference experiment[33]. The QDs are excited every 12.5 ns by two π-pulses separated by 2 ns and emit ideally two XX–X photon pairs per cycle. The photons are then guided to an unbalanced Mach–Zehnder interferometer equipped with 2 ns delay and are let to interfere at the beam splitter (BS) in co- and cross-polarized configuration.

An example of the measurements for the XX are shown in Fig. 2d. For cross-polarized photons, the three central peaks around 0 time delay show the same amplitude, as expected for distinguishable photons. Contrarily, the strong suppression of the central peak for co-polarized photons clearly indicates the occurrence of two-photon interference and pinpoints to a large indistinguishability. A similar behaviour is observed for X photons, as shown in Fig. 2e. By Lorentzian fitting (see Supplementary Fig. 1 and Supplementary Note 1) and by taking into account the imperfections of the BS[12,34–36], we estimate visibilities of two-photon interference of $0.86 \pm 0.09$ and $0.93 \pm 0.07$ for X and XX photons, respectively (for details on the calculations, see Supplementary Note 1 and Supplementary Table 1). These values are among the highest reported so far for QDs driven under two-photon excitation and are very close to the values achieved for near-optimal single-photon sources[35,37]. While the slightly lower value of visibility of the X can be explained by the time jitter introduced by the XX radiative decay, we emphasize that this is not a general feature of our QDs.

We performed the same measurements in eight additional QDs, and for some of them, we report the relevant results in Fig. 2e. Large visibility (exceeding 70%) of two-photon interference has been found in 80% of the QDs we measured, for both X and XX photons. The observed fluctuations can be explained by the different charge-noise configuration in each specific QD, as also reported in the literature[35]. It is a well-known problem that spectral diffusion induced by charge fluctuations can lead to a significant drop in the interference visibility at an excitation pulse separation >2 ns[38]. For this reason, we compared the visibility at 2 and 12.5 ns pulse separation (see Supplementary Fig. 2) and observed a drop of about 30% at a longer separation time (for details, see Supplementary Note 2). This effect would reduce the maximum possible two-photon interference visibility of two remote QDs[36], which is a major issue for the realization of a quantum repeater. A possible solution to this problem is to apply an electric field across the QD so as to stabilize the charge environment[35,39].

**Strongly entangled photons**. Finally, we complete our study by demonstrating that our QDs emit photon pairs with a high degree of entanglement. Here we take advantage of two features of our QDs: First, their high structural symmetry (see Fig. 1a and 1c), which results in an extremely small average fine-structure splitting (FSS) between the two X states[25] and second the short radiative decay time of X ($T_1$ around 250 ps, see Supplementary Fig. 3 and Supplementary Note 3). It is indeed well known that only if the FSS is smaller or comparable to the radiative linewidth of the transitions, a large degree of entanglement can be measured in a time integrated experiment[40]. The used QDs have an average FSS of $3.9 \pm 1.8\,\mu\mathrm{eV}$, as reported earlier[25]. The FSS of QD1–3 are, respectively, $6.5 \pm 0.5\,\mu\mathrm{eV}$, $1.3 \pm 0.5\,\mu\mathrm{eV}$ and $1.2 \pm 0.5\,\mu\mathrm{eV}$. Therefore, we do not expect a high degree of entanglement for QD1 (see Supplementary Fig. 4) and only QD2 and QD3 are retained for the following study.

We start out with QD2 and perform a full quantum-state tomography[41] to reconstruct the two-photon density matrix (see Fig. 3a) with the aid of a maximal likelihood method[41]. In order to quantify the degree of entanglement, we extract the concurrence $\zeta$ from the density matrix and find $\zeta = 0.84 \pm 0.05$. Closer inspection of the density matrix reveals that the eigenstate associated with the largest eigenvalue ($0.92 \pm 0.03$) is not exactly the expected Bell state $|\psi^+\rangle = 1/\sqrt{2}(|H_{XX}\rangle|H_X\rangle + |V_{XX}\rangle|V_X\rangle)$[40], but rather $|\psi_r^+\rangle \approx 1/\sqrt{2}(|H_{XX}\rangle|H_X\rangle + e^{i\alpha\pi}|V_{XX}\rangle|V_X\rangle)$, with $\alpha = 0.12$. The phase is most probably induced by the residual

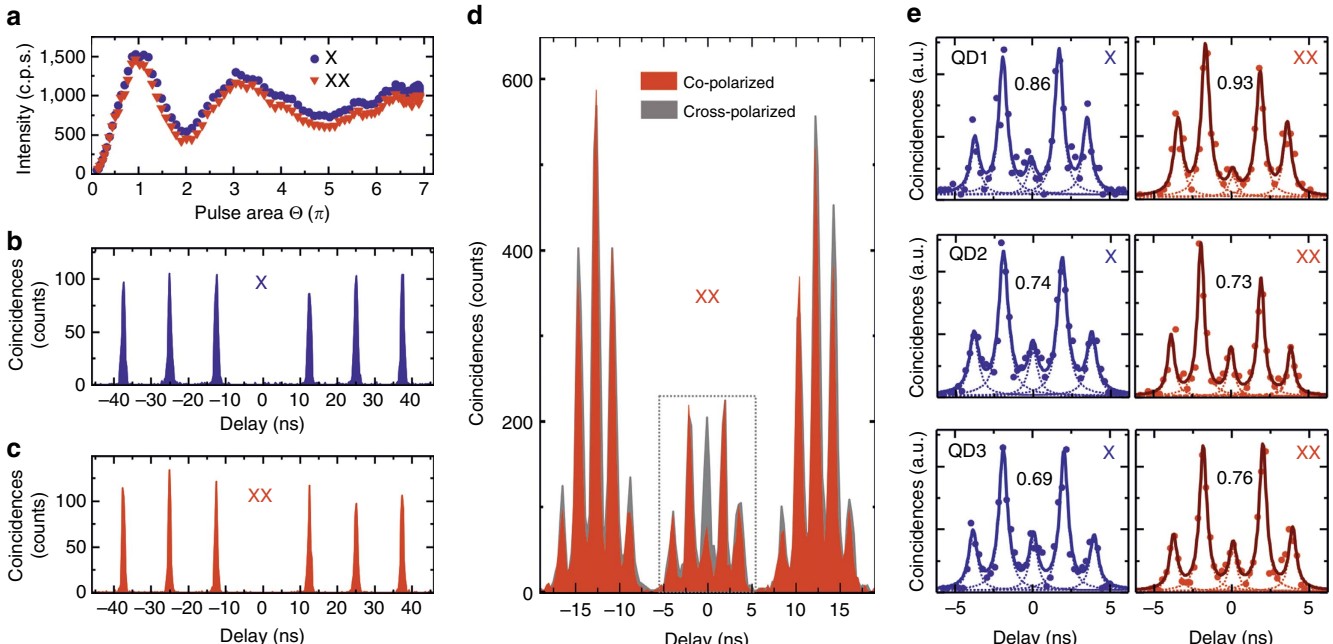

**Figure 2 | Single-photon purity and two-photon interference. (a)** Integrated intensities of the excition (X) (blue) and biexciton (XX) (red) versus the square root of the pump power under resonant excitation of a representative quantum dot (QD). The abscissa is normalized in $\pi$-units with respect to the $\pi$-pulse of the XX state. The X as well as the XX show well-defined Rabi oscillations. **(b)** Auto-correlation measurements of X and **(c)** XX photons for a representative QD. The measurements show strong antibunching for both lines. **(d)** Two-photon interference with co-polarized (red) and cross-polarized (grey) XX photons. **(e)** Same as in **d** for three different QDs around zero time delay (see dotted line box in **d**). The data for both the X and XX co-polarized photons are reported. The dashed lines are Lorentzian fits of the peaks. The solid lines show the total sum of the single fits. The values of the visibility are also given in each panel.

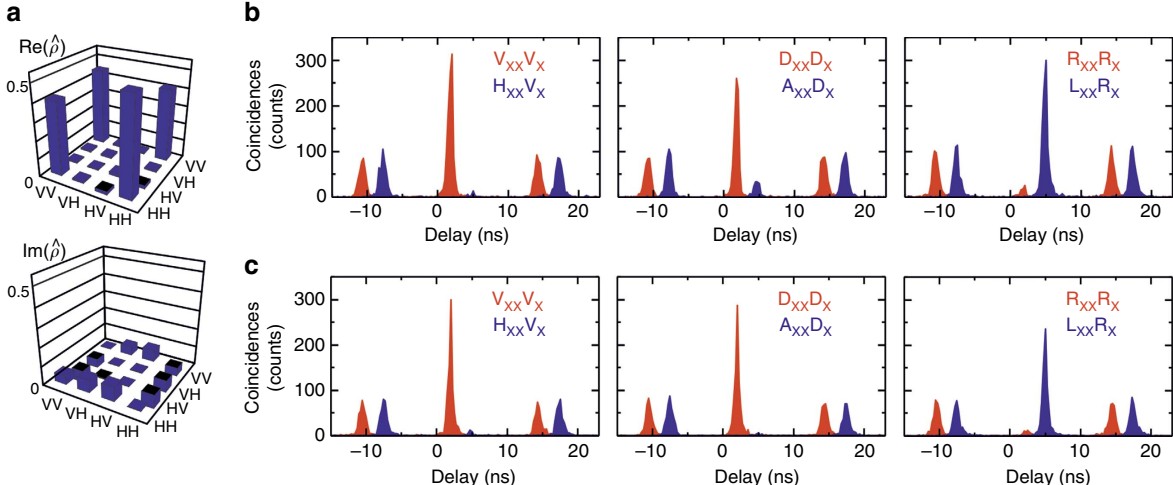

**Figure 3 | Entangled photons from GaAs quantum dots. (a)** Real and imaginary part of the two-photon density matrix reconstructed via quantum state tomography on quantum dot QD2. The matrix has been calculated out of biexciton–exciton (XX–X) cross-correlation measurements in 16 polarization settings and using a maximum likelihood method. **(b)** XX–X cross-correlation measurements under resonant two-photon excitation for QD2 and **(c)** QD3. $V_{XX,X}$ ($H_{XX,X}$), $D_{XX,X}$ ($A_{XX,X}$) and $R_{XX,X}$ ($L_{XX,X}$) indicate vertical (horizontal), diagonal (antidiagonal) and circularly right (circularly left) polarized photons. The graphs for co-polarized (red) and cross-polarized (blue) photons are temporally shifted by 3 ns for clarity.

FSS and/or by the measurement setup and in particular by the presence of background photons of the resonant laser that could not be suppressed completely during quantum state tomography (see Supplementary Fig. 5).

In order to demonstrate that the high entanglement degree is not a feature of one specific QD, we extend the study to QD3. In this case, we took special care in rejecting the resonant laser and

we extracted the fidelity to the expected Bell state via

$$f = \frac{1 + C_{\text{linear}} + C_{\text{diagonal}} - C_{\text{circular}}}{4}, \quad (1)$$

where $C$ are the correlation visibilities in the linear, diagonal and circular bases (see Supplementary Note 5). In fact, for QD2 (see Fig. 3b) we confirmed that the value of the fidelity obtained with

this formula (0.88 ± 0.01) is very close to the one extracted via full reconstruction of the density matrix (0.92 ± 0.03) and, since $\alpha$ is small, to the largest eigenvalue (0.92 ± 0.03). We have therefore used the six measurements reported in Fig. 3c to extract the fidelity from equation (1), which is found to be 0.94 ± 0.01 (for details, see Supplementary Table 2). This value—which is higher than the one measured for QD2 due to improved rejection of the laser (see Supplementary Fig. 5 for comparison of the $g^{(2)}(0)$ values)—is very close to the maximum achievable (see Supplementary Fig. 4). It is worth mentioning that we obtain this result without any temporal/spectral filtering of the emitted photons and without the aid of the Purcell effect[37], which can in principle be used to shorten the X radiative decay time even further, and to improve the level of entanglement and indistinghuishability to the maximum. The high degree of entanglement allows for violation of Bell's inequality. Using the following expressions:

$$S_{RC} = \sqrt{2}(C_{linear} - C_{circular}) \le 2 \qquad (2)$$

$$S_{DC} = \sqrt{2}(C_{diagonal} - C_{circular}) \le 2 \qquad (3)$$

$$S_{RD} = \sqrt{2}(C_{linear} + C_{diagonal}) \le 2 \qquad (4)$$

we obtain $S_{RC} = 2.52 \pm 0.02$, $S_{DC} = 2.59 \pm 0.01$ and $S_{RD} = 2.64 \pm 0.01$. The latter parameter shows violation of Bell's inequality by >60 standard deviations, a result that we achieve without any temporal or spectral filtering.

The measurements of Fig. 3c readily suggest the origin of this high level of entanglement. In fact, an almost perfect antibunching is observed for cross-linear, cross-diagonal and co-circular base. This result clearly indicates a strong suppression of X spin-flip processes[42] and of recapture, a result which is a consequence of the resonant excitation scheme used. Moreover, the fact that the correlation visibility is lower in the circular than in the linear/diagonal base suggests that the X dephasing induced by the hyperfine interaction is quite limited in these QDs[19]. In fact, a theoretical model[42], which takes into account the effect of the Overhauser field on dephasing of the intermediate X states (see Supplementary Note 4) shows that this is exactly the reason why our GaAs QDs show such a high level of entanglement. The same model can also be used to highlight that the small deviation from unity is mainly ascribable to the non-zero values of the FSS. More specifically, we estimate that in GaAs QDs with suppressed FSS—reachable, for example via three-axial strain engineering[24]—the fidelity can reach values as high as 0.99, roughly 10% higher than what can be theoretically achieved with InGaAs QDs with FSS = 0 with the same X decay time (see Supplementary Note 4).

## Discussion

Our results show that the GaAs/AlGaAs material system, which has led to breakthrough discoveries in condensed matter physics and optoelectronics (such as the fractional quantum Hall effect[43]) has also a great potential as entanglement resources for long distance quantum communication and in particular for quantum repeaters[1,3]. In contrast to parametric downconverted sources of non-classical light, low temperatures are usually required to use them as highly indistinguishable entangled photon sources[38], but this is not really a hurdle considering recent advances in the fabrication of compact, inexpensive and low-vibration cryocoolers[44]. Beside the near-optimal level of entanglement and indistinguishability of the emitted photons, our QDs emit in the spectral range of absorption resonances of rubidium atoms, a system which has proven its potential as long-lived and efficient quantum memory[45]. Moreover, the availability of downconversion techniques[46] in combination with methods to

convert polarization entanglement into time-bin entanglement[47,48] make our QDs suited for quantum communication at telecom wavelengths. A key ingredient to achieve high single-photon purity, high two-photon interference and high level of entanglement is the two-photon resonant excitation. This excitation method requires stabilization of the laser energy and power, as small fluctuations in these parameters dramatically affect the biexciton preparation fidelity, thus making the scheme less attractive for the envisioned applications. However, we would like to point out that two alternative excitation regimes exist: phonon-assisted population[49,50] and rapid adiabatic passage[51,52]. While the first method uses an incoherent population of the XX via acoustic phonons, the second one allows the XX state to be coherently populated via chirped laser pulses. Both schemes are more robust against small variations of the energy of the laser as well as fluctuations of the laser pump power. In addition, to really use these QDs in real-life applications, such as quantum repeater networks, the next step is to integrate them in photonic structures for boosting the flux of the XX and X photons and to gain control over the energy of the emitted photons using external perturbations[24,53–55]. We envisage that the implementation of these tasks in the GaAs QD system will soon lead to the fabrication of an ideal source of entangled photons.

## Methods

**Sample growth.** The sample is grown by solid state molecular beam epitaxy. A GaAs (001) substrate is overgrown with a 270 nm buffer layer, on which three pairs of alternating $Al_{0.2}Ga_{0.8}As$/AlAs (with a thickness of 56 and 65 nm) are deposited. On top of this layer structure, which forms a low reflectivity back mirror of a planar cavity, a 120 nm thick layer of $Ga_{0.6}Al_{0.4}As$ is grown. This layer is hosting the nanoholes, which are fabricated at 600 °C using aluminum droplet etching[25]. The QDs are then obtained by depositing 2.5 nm of GaAs that fills the nanoholes during a 2 min annealing process. This layer is subsequently capped with 120 nm of $Al_{0.4}Ga_{0.6}As$ acting as top barrier. Finally, two pairs of $Al_{0.2}Ga_{0.8}As$/AlAs layers were grown to complete the cavity. Due to the limited number of pairs used for the distributed Bragg reflector mirrors, we do not expect any appreciable Purcell effect but only a slight enhancement of the light extraction efficiency.

**Measurement setup.** All the measurements are performed at a sample temperature of 5 K in a helium-flow cryostat. For non-resonant excitation, a 488 nm continuous wave laser is used. The resonant two-photon excitation is performed with a titanium sapphire (TiSa) femtosecond laser featuring a bandwidth of 100 fs and a repetition rate of 80 MHz. The TiSa is shaped by a 4f-pulse-shaper setup into 9 ps pulses. The excitation lasers are focussed via an objective with a numerical aperture of 0.42 onto the sample, on which a hyperspherical solid immersion lens, made of zirconia, is placed. We estimate that the extraction efficiency of our device (with solid immersion lens on top of the sample) is around 1%. The same objective is used for the collection of the photoluminescence signal.

To reduce the back scattered laser light in resonant excitation, tunable notch filters with a bandwidth of 0.4 nm are placed in the beam path. Further stray light reduction is achieved by coupling the signal into polarization maintaining single-mode fibres placed after a non-polarizing BS. Depending on the QD, the addition of a continuous wave laser at extremely low (not measurable) power to the resonant laser increases the X and XX count rate. This non-resonant laser—which does not give rise to background photons—most probably fills traps in the QD surrounding.

After the fibres, light is spectrally filtered by one or two independent spectrometers and detected by a charge-coupled device camera or—during photon correlation spectroscopy—by avalanche photodiodes connected to the correlation electronics. The temporal resolution of these detectors is about 500 ps.

The measurements of the FSS are carried out with a rotating $\lambda/2$ wave plate in front of a fixed polarizer at the entrance of a double spectrometer. Using Lorentzian fitting of both the X and XX lines, the FSS can be measured with sub-$\mu$eV resolution. For quantum state tomography, polarizers and properly oriented $\lambda/2$ and $\lambda/4$ wave plates are inserted in the X and XX path, before the two fibre inputs.

For the two-photon interference experiment, the femtosecond laser pulses are sent into an unbalanced Mach–Zehnder interferometer with a 2 ns delay to create two pulses every 12.5 ns. Both laser pulses are shaped with the 4f-pulse-shaper described above. A polarization maintaining fibre BS is used for two-photon interference of QD photons. The measurements for co-linear and cross-linear configuration are obtained by placing polarizers before the input of the fibre BS. The latter is characterized with the same pulse-shaped laser used to excite the QDs

and shows a mode overlap $(1-\varepsilon)=0.96\pm0.01$, while the reflection and transmission coefficient are found to be $0.52\pm0.005$ and $0.48\pm0.005$, respectively.

**Data availability.** The data that support the findings of this study are available from the corresponding authors upon request.

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

## Acknowledgements

This work was financially supported by European Union's Horizon 2020 research and innovation programme (SPQRel grant agreement no. 679183), European Union Seventh Framework Programme (FP7/2007–2013) under grant agreement no. 601126 (HANAS) and Austrian Science Fund (FWF): P 29603. We thank Dr Javier Martín-Sánchez, Dr Klaus Jöns, Florian Sipek, Mathias Gartner and Maximilian Prilmüller for fruitful discussions and for help at the very early state of the project.

## Author contributions

D.H. and M.R. performed the measurements with the help from R.T., D.H. performed the data analysis with the help from M.R., J.S.W. and R.T. Y.H. grew the sample with support of A.R. and O.G.S. H.H. and M.R. characterized the sample. D.H. and R.T. wrote the manuscript with the help from all the authors. R.T. and A.R. conceived the project. R.T. designed the experiments and supervised the project.

## Additional information

**Competing interests:** The authors declare no competing financial interests.

**Publisher's note**: 

