## [Peer Review File · Nature Communications]

Reviewers' comments:

Reviewer #1 (Remarks to the Author):

In my opinion, the authors carefully addressed all the referees' comments in their rebuttal letter. Their argumentation is convincing and they provide a revised manuscript which takes into account all these comments. Therefore, I still think that this work deserves to be published.

Without minimizing the high scientific interest of this work, due to the lack of novelty regarding its "material" aspect (as this was raised in the discussion), an eventual publication in Nature Communications rather than in [redacted] appears indeed more suitable.

Reviewer #2 (Remarks to the Author):

Second report on the manuscript 117417_1 (54173) entitled:

Highly indistinguishable and strongly entangled photons from symmetric GaAs quantum dots

by D. Huber, M. Reindl, Y. Huo, H. Huang, J. S. Wildmann, O. G. Schmidt, A. Rastelli, and R. Trotta.

Clearly, the main originality of the paper relies on the simultaneous measurement of the photon indiscernibility emitted by a GaAs quantum dot and the degree of polarization entanglement of the photons emitted through the biexciton cascade towards the ground state. In addition, the efficient preparation of biexciton with resonant two-photon absorption minimizes the charge fluctuation in the QD environment, leading to some improved indiscernibility and entanglement degrees with respect to the present state of the art (this without any post selection, nor complicated exciton fine structure splitting compensation).

The authors responded convincingly to most of the recommendation and criticisms of the referees, so that the manuscript presented here has been significantly improved with respect to the initial version. It should thus deserve publication in Nature Communication provided that they take into account the following points:

1. Figure 2 has been improved, but still the horizontal axis legend is somehow ambiguous: as a fact I understand that the definition of a “ π ” pulse is related here to the biexciton one. It does not match with the “ π ” pulse for excitons, which is typically twice lower (see e.g. Stufler et al. ref. 34 of the main text - in particular fig. 4-). The fact that the biexciton and exciton intensity oscillations have the same period with respect to the square root of the excitation power can be understood here as due to the fact that the excitons are only emitted by spontaneous emission from the photo-generated biexciton. Hence the present axis legend could be misleading for the reader, unless the author clarify its meaning, in the main text or in the figure legend.

2. Although the strategy adopted by the authors is validated by their results, still open technological questions remain for the practical implementation of these dots in a quantum network:

- First, the temperature should be strictly maintained below 10 K (cf. ref 16 of supplement), while parametric crystals can easily operate at room temperature.
- Although the proximity of the Rb transition may be useful for photon storage, the emission wavelength of GaAs QDs is far from the optimum attenuation windows of silica optical fibres currently in use in optical telecommunications. In addition, polarization maintaining fibres should also be used.

The authors should comment about these points in their discussion.

3. As mentioned by another referee and recognized by the authors the high fidelity biexciton preparation method, although efficient here, is not robust with respect to the excitation laser detuning and amplitude fluctuations. While the solution invoked on line 89 of the supplement - which rely on efficient phonon-assisted generation of biexcitons as predicted in ref. 11 - is a possible improvement, the author should mention the alternative of rapid adiabatic passage towards the biexciton using chirped pulses, which is also robust with respect to frequency detuning and amplitude fluctuations of the excitation laser, provided that the regime of dynamical decoupling with the phonon bath is reached. This was theoretically predicted e.g. in Debnath et al. [Phys. Rev. B **88**, 201305(R)

(2013)], and recently demonstrated experimentally (see: T. Kaldewey et al., arXiv: 1701.01371v1 and references therein).

I think it is necessary that the authors include this alternative method and corresponding references in the discussion about robustness of biexciton preparation.

4. Minor detail: in Fig. 2 legend (main text), instead of “(see grey box in c)” one should read rather e.g.: “(see dotted line box in c)”, as there is no grey box in the figure.

To conclude, this work displays some real scientific interest both for the semiconductor quantum dot and quantum optics communities. I think it could be published by Nature Communications provided the authors made the improvements mentioned above.

Reviewer #3 (Remarks to the Author):

In their revised manuscript and in the response letter, D. Huber et al. addressed my technical remarks in a comprehensive and satisfactory way. Some comments on their replies:

- The authors commented on the issue of extraction efficiency and mention 5% for their device in the manuscript. It is clear that this value needs to be significantly improved for devices suitable for implementing a quantum repeater network.

- I appreciate the authors performed additional HOM measurements for larger pulse separation (12.5 ns in addition to 2 ns). These measurements show a significant drop in the visibility of about 30% which is smaller than 44% observed under p-shell excitation by Thoma et al.. Still, the decrease is more significant than I would have expected for an experiment under resonant two-photon excitation – having an almost pulse-separation independent visibility under strict resonant excitation (see H. Wang et al. Phys. Rev. Lett. 116, 213601 (2016)) in mind. In this regard, it should clearly be stated in the abstract that $V = 0.93$ was obtained at a pulse separation of 2 ns (in best case with mentioning also that V drops by 30% for larger pulse separation).

- Regarding piezo-tuning the authors argue that this scheme was not in the scope of the present work. Still I wonder if it is feasible technologically in the present material system (the work cited in the rebuttal by R. Trotta et al., Nature Comm. 7, 10375 (2016) was performed on InGaAs QDs), i.e. are suitable sacrificial-layer materials available to allow for the flip-chip process necessary for transferring a thin semiconductor membrane to the piezo actuator? It would be good to briefly address this question.

What remains is essentially the question about the suitability of the present work for publication in Nature Communications. I understand that the goal of the present work is to demonstrate the high potential of the symmetric GaAs QDs to act as state-of-the-art emitters of entangled and indistinguishable photon pairs – a combination which is difficult to archive with as-grown InGaAs QDs. Indeed, the achieved numbers $g(2)(0)=0.002$, $V = 0.93$, and F and high entanglement fidelity $F = 0.94$ clearly demonstrate the high potential of symmetric GaAs QDs for the proposed application in quantum technology. Still, I share the opinion of reviewer 2 saying that the work is rather incremental as it combines many aspects which have been demonstrated previously for the present QDs (e.g. small FSS, see Y. Huo, et al., App. Phy. Lett. 102, 152105 (2013)) or for InGaAs QDs (e.g. the combination of high V and F under two photon excitation, see M. Müller et al., Nature Photon. 8, 224–228 (2014)). Moreover, very similar results (except of high indistinguishability) are reported in arxiv:1611.03717.

Additionally, as already indicated in my previous report, I think real progress with the necessary impact to warrant publication in a Nature journal would be achieved by succeeding in a two source Hong-Ou-Mandel experiment in combination with QDs emitting entangled photon pairs with high fidelity. This important result was actually achieved by the authors (via phonon assisted TPE) as reported in arxiv:1701.07812. I think that these results outdate the present ones which are (only) a prerequisite of new experiment.

Thus, in my opinion and in spite of the excellent technical quality of the work, the present manuscript by D. Huber et al. does not include the novelty, innovation and impact to warrant publication Nature Communications.

Reviewer#1 (Remarks to the Author):

In my opinion, the authors carefully addressed all the referees' comments in their rebuttal letter. Their argumentation is convincing and they provide a revised manuscript which takes into account all these comments. Therefore, I still think that this work deserves to be published. Without minimizing the high scientific interest of this work, due to the lack of novelty regarding its "material" aspect (as this was raised in the discussion), an eventual publication in Nature Communications rather than in [redacted] appears indeed more suitable.

We thank the Reviewer once again for her/his very positive judgment on our work and we are very glad that the Reviewer has grasped its originality and importance.

Reviewer #2 (Remarks to the Author):

Clearly, the main originality of the paper relies on the simultaneous measurement of the photon indiscernibility emitted by a GaAs quantum dot and the degree of polarization entanglement of the photons emitted through the biexciton cascade towards the ground state. In addition, the efficient preparation of biexciton with resonant two-photon absorption minimizes the charge fluctuation in the QD environment, leading to some improved indiscernibility and entanglement degrees with respect to the present state of the art (this without any post selection, nor complicated exciton fine structure splitting compensation). The authors responded convincingly to most of the recommendation and criticisms of the referees, so that the manuscript presented here has been significantly improved with respect to the initial version. It should thus deserve publication in Nature Communication provided that they take into account the following points:

First of all, we would like to thank the Reviewer for providing comments that helped improving our manuscript. We are very glad that she/he finds that our work deserves publication in Nature Communications. Below, we address one by one the points raised by the Referee.

1. Figure 2 has been improved, but still the horizontal axis legend is somehow ambiguous: as a fact I understand that the definition of a "pi" pulse is related here to the biexciton one. It does not match with the "pi" pulse for excitons, which is typically twice lower (see e.g. Stufler et al. ref. 34 of the main text - in particular fig. 4-). The fact that the biexciton and exciton intensity oscillations have the same period with respect to the square root of the excitation power can be understood here as due to the fact that the excitons are only emitted by spontaneous emission from the photo-generated biexciton. Hence the present axis legend could be misleading for the reader, unless the author clarify its meaning, in the main text or in the figure legend.

We thank the Reviewer for this comment. We completely agree that pi-pulse condition is referred to the biexciton, and that the exciton shows the same intensity oscillations because it is populated via the spontaneous emission of the biexciton state. We have thus added a note in the main text: "We note that while the XX state and the crystal ground state are coherently coupled by the laser field, X emission stems from spontaneous decay of the XX state. Therefore the X population reflects that of the XX state and shows Rabi oscillations with the same periodicity." We also agree that the figure could generate some confusion and,

following her/his suggestions, we have modified the figure caption by explicitly stating that “The abscissa is normalized in pi-units with respect to the pi-pulse of the XX state”.

2. Although the strategy adopted by the authors is validated by their results, still open technological questions remain for the practical implementation of these dots in a quantum network:

- First, the temperature should be strictly maintained below 10 K (cf. ref 16 of supplement), while parametric crystals can easily operate at room temperature.

- Although the proximity of the Rb transition may be useful for photon storage, the emission wavelength of GaAs QDs is far from the optimum attenuation windows of silica optical fibres currently in use in optical telecommunications. In addition, polarization maintaining fibres should also be used. The authors should comment about these points in their discussion.

We thank the Reviewer for these comments.

- We agree with the Reviewer that, in contrast to parametric-down converters, QDs usually require low temperatures to achieve the best performances in terms of entanglement, purity, and indistinguishability of the emitted photons. However, we would like to point out that temperatures around 10 K are nowadays not a technological issue. Besides available closed-cycles system that can easily achieve sub-10 K temperature without liquid helium, there has been substantial progress in the development of inexpensive, compact and low-vibration Stirling cryocoolers, see for example A. Schlehahn et al. Rev. Sci. Instr. 86, 013113 (2015). While this work has demonstrated $T < 30$ K (sufficient for entangled photon generation, see R. Hafenbrack et al., New. J. Phys. 9, 315 (2007)), there is little doubt that even more compact and inexpensive solution for T around 10 K will be realized in the near future. **Following the comment of the Reviewer, we have mentioned the progress in the fabrication of inexpensive cryocoolers in the revised version of the manuscript and we have added a new reference (number 44).**

- For what concerns the emission wavelength of our GaAs QDs, we agree with the Reviewer that the **direct use** of photons from GaAs QDs is not suitable for current optical telecommunications. However, there exists an exciting possibility to overcome this problem. It is indeed possible to down-convert near-infrared photons to telecom wavelengths using periodically poled lithium niobate (PPLN) waveguides. This approach, which has shown to preserve the single photon character and the shape of the photon wavepacket (see for example Ates, S. et al., Phys. Rev. Lett. 109, 147405 (2012)), is particularly suitable for our GaAs QDs, as the down-converted photons would have a wavelength that matches perfectly the telecommunication C-band. Also the use of polarization-maintaining fibers can be avoided. In fact, it is possible to convert polarization entanglement into time-bin entanglement, the latter being almost immune to decoherence processing occurring during propagation in standard optical fibers. Most notably, the two approaches (polarization to time bin conversion and the use of PPLN waveguides) have been demonstrated recently in L. Yu et al., Nature Comm. 6, 8955 (2015) for InGaAs QDs. For future applications, it would be extremely interesting to adapt this concept to GaAs QDs. **Following the comment of the Reviewer, we have mentioned this possibility in the revised version of the manuscript and we have added new references (number 46, 47).**

3. As mentioned by another referee and recognized by the authors the high fidelity biexciton preparation method, although efficient here, is not robust with respect to the excitation laser detuning and amplitude fluctuations. While the solution invoked on line 89 of the supplement - which rely on efficient phonon-assisted generation of biexcitons as predicted in ref. 11 - is a possible improvement, the author should mention the alternative of rapid adiabatic passage towards the biexciton using chirped pulses, which is also robust with respect to frequency detuning and amplitude fluctuations of the excitation laser, provided that the regime of dynamical decoupling with the phonon bath is reached. This was theoretically predicted e.g. in Debnath et al. [Phys. Rev. B 88, 201305(R) 2 (2013)], and recently demonstrated experimentally (see: T. Kaldewey et al., arXiv: 1701.01371v1 and references therein). I think it is necessary that the authors include this alternative method and corresponding references in the discussion about robustness of biexciton preparation.

We thank the Reviewer for this comment, and we agree that “rapid adiabatic passage” is also a robust solution against fluctuation of the laser detuning and power. Following the suggestion of the Reviewer, we have commented on this point in the revised version of the manuscript and we have added the two references She/He has suggested (see reference 49 - 52). Further, we decided to move this important discussion from the supplementary to the discussion paragraph of the main text.

4. Minor detail: in Fig. 2 legend (main text), instead of “(see grey box in c)” one should read rather e.g.: “(see dotted line box in c)”, as there is no grey box in the figure.

We thank the Reviewer for this comment, as we indeed made a mistake. Following her/his suggestion, we have corrected the caption in the revised version of the manuscript.

To conclude, this work displays some real scientific interest both for the semiconductor quantum dot and quantum optics communities. I think it could be published by Nature Communications provided the authors made the improvements mentioned above.

Reviewer #3 (Remarks to the Author):

In their revised manuscript and in the response letter, D. Huber et al. addressed my technical remarks in a comprehensive and satisfactory way. Some comments on their replies:

First of all, we would like to thank the Reviewer for providing comments that helped improving our manuscript. Below, we address one by one the points raised by the Referee.

- The authors commented on the issue of extraction efficiency and mention 5% for their device in the manuscript. It is clear that this value needs to be significantly improved for devices suitable for implementing a quantum repeater network.

We agree with the Reviewer that this value (meanwhile we have recalculated the value of the extraction efficiency, which is about 1%) needs to be improved for implementing a quantum repeater network based on QDs. This is feasible in the near future by using GaAs micro-lenses, which are supposed to increase the extraction efficiency up to the maximum level (a value of $\sim 90\%$ has been predicted in M Gschrey et al, Nature Communications 6, 1662 (2014)). We have already commented on this point in the previous version of the manuscript. However, following the Reviewer's comment we have explicitly referred to quantum repeater networks in the discussion part of the revised version of the manuscript.

- I appreciate the authors performed additional HOM measurements for larger pulse separation (12.5 ns in addition to 2 ns). These measurements show a significant drop in the visibility of about 30% which is smaller than 44% observed under p-shell excitation by Thoma et al.. Still, the decrease is more significant than I would have expected for an experiment under resonant two-photon excitation – having an almost pulse-separation independent visibility under strict resonant excitation (see H. Wang et al. Phys. Rev. Lett. 116, 213601 (2016)) in mind. In this regard, it should clearly be stated in the abstract that $V = 0.93$ was obtained at a pulse separation of 2 ns (in best case with mentioning also that V drops by 30% for larger pulse separation).

We are glad that the Reviewer appreciated our additional experimental efforts to answer her/his concerns. We also agree that the value of the pulse separation is an important parameter when giving the visibilities of two photon interference. Therefore, following the suggestion of the Reviewer we have included the 2 ns pulse separation in the abstract of the revised version of the manuscript. Concerning the drop of visibility at longer pulse separations, we believe that this detail is not so relevant to be included in the abstract, especially considering the length-limit we have to respect to comply with the format of Nature Communications.

- Regarding piezo-tuning the authors argue that this scheme was not in the scope of the present work. Still I wonder if it is feasible technologically in the present material system (the work cited in the rebuttal by R. Trotta et al., Nature Comm. 7, 10375 (2016) was performed on InGaAs QDs), i.e. are suitable sacrificial-layer materials available to allow for the flip-chip process necessary for transferring a thin semiconductor membrane to the piezo actuator? It would be good to briefly address this question.

We thank the Reviewer for this comment. The strain-tuning technique can be easily used for GaAs QDs using the same sacrificial layers (Al-rich AlGaAs) used for InGaAs QDs. In fact, we have fabricated strain-tunable devices containing GaAs QDs (using sacrificial layer and flip chip bonding) already in 2011 (see Kumar S. et al., Appl. Phys. Lett. 99, 161118 (2011)) and also very recently in a LED-like nanomembrane (see Huang H. et al., arXiv:1602.02122 (2016)). Following the comment of the Reviewer we have inserted two new references (see number 54 and 55) in the discussion of the strain-tuning part.

What remains is essentially the question about the suitability of the present work for publication in Nature Communications. I understand that the goal of the present work is to

*demonstrate the high potential of the symmetric GaAs QDs to act as state-of-the-art emitters of entangled and indistinguishable photon pairs – a combination which is difficult to archive with as-grown InGaAs QDs. Indeed, the achieved numbers $g(2)(0)=0.002$, $V = 0.93$, and F and high entanglement fidelity $F = 0.94$ clearly demonstrate the high potential of symmetric GaAs QDs for the proposed application in quantum technology. Still, I share the opinion of reviewer 2 saying that the work is rather incremental as it combines many aspects which have been demonstrated previously for the present QDs (e.g. small FSS, see Y. Huo, et al., *App. Phys. Lett.* 102, 152105 (2013)) or for InGaAs QDs (e.g. the combination of high V and F under two photon excitation, see M. Müller et al., *Nature Photon.* 8, 224–228 (2014)).*

We disagree with the Reviewer on this point, as our manuscript contains truly novel and important results. In fact, we demonstrate for the first time that GaAs QDs can generate indistinguishable entangled photons on demand. We would like to stress that the indistinguishability of photons from GaAs QDs has never been reported in the literature. Moreover, the values of entanglement fidelity and visibility of two-photon interference we demonstrate in our work clearly outperform those achievable in InGaAs QDs under similar conditions (no use of Purcell effect etc.). This is an extremely important message if one considers that the vast majority of quantum optical studies performed on QDs are focusing on the InGaAs system. Here, we are proving that GaAs QDs are more promising for future applications in quantum information processing. As also Reviewer 2 eventually realized, our “work displays some real scientific interest both for the semiconductor quantum dot and quantum optics communities”.

Moreover, very similar results (except of high indistinguishability) are reported in arxiv:1611.03717.

Here, we would like to emphasize two points:

1) Our work was posted on Arxiv (arXiv:1610.06889) almost one month before the work mentioned by the Referee. The two works have an overlapping co-author who has never informed us (the corresponding authors writing this letter) about the work, which appeared in arxiv:1611.03717. From informal communication between A.Rastelli and the corresponding author of arxiv:1611.03717 (after posting of our arXiv), we concluded that the two works proceeded independently of each other and relied on different samples. We thus not see how arxiv:1611.03717 can reduce the value of our work.

2) As the Reviewer has correctly pointed out, the work in arxiv:1611.03717 does not contain a study on the indistinguishability of the emitted photons. This is however a substantial difference. In fact, almost all the existing quantum communication and information protocols rely on the indistinguishability of the emitted photons. It is exactly the high indistinguishability combined with the unprecedented degree of entanglement the main message of our work. As stated by Reviewer 2, “the main originality of the paper relies on the simultaneous measurement of the photon indiscernibility emitted by a GaAs quantum dot and the degree of polarization entanglement”. This is exactly the key point that makes our manuscript of superior quality.

Additionally, as already indicated in my previous report, I think real progress with the necessary impact to warrant publication in a Nature journal would be achieved by succeeding in a two source Hong-Ou-Mandel experiment in combination with QDs emitting entangled photon pairs with high fidelity. This important result was actually achieved by the authors (via phonon assisted TPE) as reported in arxiv:1701.07812. I think that these results outdate the present ones which are (only) a prerequisite of new experiment.

While we are glad that the Referee finds our recent work (which is however not published) on two QDs important, we strongly believe that this does not lower the impact of our current manuscript: First, the results shown in arxiv:1701.07812 focus on a completely different excitation scheme and physics (phonon-assisted TPE). Secondly, the value of entanglement and visibility of two-photon inference are lower than what we achieve here. Third, the results shown in arxiv:1701.07812 demonstrate once again the superior quality of the GaAs QDs, a message that we would like to convey for the first time with the present manuscript. Following the comment of the Reviewer, we have now included reference to arxiv:1701.07812 in the revised version of the manuscript (see number 50).

Thus, in my opinion and in spite of the excellent technical quality of the work, the present manuscript by D. Huber et al. does not include the novelty, innovation and impact to warrant publication Nature Communications.

List of changes:

Main Paper:

- 1. Corresponding author list has been reduced to 3 Authors according to Nature Communications Manuscript checklist.**
- 2. Abstract has been shortened to fulfill the Nature Communications Manuscript checklist. Further, the references have been removed.**
- 3. Results: new sub headings have been introduced according to Nature Communications Manuscript checklist**
- 4. Line 79-81: Sentence has been added to clarify the resonant excitation of the XX state according to the suggestion of reviewer 2.**
- 5. Fig 2: Figure caption has been changed according to suggestion of reviewer 2.**
- 6. Line 157-160: Discussion has been expanded according to the suggestions of reviewer 2 about the low temperature requirement for quantum dots**
- 7. Line 162-164: Discussion has been expanded according to the suggestions of reviewer 2 about the wavelength matching in fibers**
- 8. Line 164-171: The discussion about the instability of the resonant excitation has been moved from the supplementary material to the main text. Further, it has been expanded by a discussion about rapid adiabatic passage according to the Reviewer suggestion.**
- 9. Line 171: The text has been modified according to comment of reviewer 3 about the extraction efficiency.**
- 10. Line 191: Extraction efficiency has been recalculated**
- 11. Data availability section has been added according to Nature Communications Manuscript checklist**
- 12. Reference style has been changed according to Nature Communications Manuscript checklist**

Supplemental Material:

- 1. Corresponding author list was reduced to 3 Authors according to Nature Communications Manuscript checklist.**
- 2. Line 85: The discussion about the instability of the resonant two photon excitation has been shifted to the main paper**